# Joint influences of obesity, diabetes, and hypertension on indices of ventricular remodeling: Findings from the community-based Framingham Heart Study

Beatrice von Jeinsen[1], Ramachandran S. Vasan[1,2,3], David D. McManus[4], Gary F. Mitchell[5], Susan Cheng[6], Vanessa Xanthakis[1,2,7] *

1 Boston University's and National Heart, Lung, and Blood Institute's Framingham Heart Study, Framingham, Massachusetts, United States of America, 2 Sections of Preventive Medicine and Epidemiology, and Cardiovascular Medicine, Department of Medicine, Boston University School of Medicine, Boston, Massachusetts, United States of America, 3 Department of Epidemiology, Boston University School of Public Health, Boston, Massachusetts, United States of America, 4 Division of Cardiovascular Medicine, Department of Medicine, University of Massachusetts Medical Center, Worcester, Massachusetts, United States of America, 5 Cardiovascular Engineering, Inc, Norwood, Massachusetts, United States of America, 6 Smidt Heart Institute, Cedars-Sinai Medical Center, Los Angeles, California, United States of America, 7 Department of Biostatistics, Boston University School of Public Health, Boston, Massachusetts, United States of America

* vanessax@bu.edu

**Data Availability Statement:** The data used for this project can be found here: https://biolincc.nhlbi.nih.gov/studies/framcohort/ https://biolincc.nhlbi.nih.

## Abstract

### Introduction

Obesity, hypertension, and diabetes are independently associated with cardiac remodeling and frequently co-cluster. The conjoint and separate influences of these conditions on cardiac remodeling have not been investigated.

### Materials and methods

We evaluated 5,741 Framingham Study participants (mean age 50 years, 55% women) who underwent echocardiographic measurements of left ventricular (LV) mass (LVM), LV ejection fraction (LVEF), global longitudinal strain (GLS), mitral E/e', left atrial end-systolic (peak) dimension (LASD) and emptying fraction (LAEF). We used multivariable generalized linear models to estimate the adjusted-least square means of these measures according to cross-classified categories of body mass index (BMI; normal, overweight and obese), hypertension (yes/no), and diabetes (yes/no).

### Results

We observed statistically significant interactions of BMI category, hypertension, and diabetes with LVM, LVEF, GLS, and LAEF (p for all 3-way interactions <0.01). Overweight and obesity (compared to normal BMI), hypertension, and diabetes status were individually and conjointly associated with higher LVM and worse GLS (p<0.01 for all). We observed an increase of 34% for LVM and of 9% for GLS between individuals with a normal BMI and

gov/studies/gen3/ https://biolincc.nhlbi.nih.gov/
studies/framoffspring/ https://biolincc.nhlbi.
nih.gov/studies/fhs/.

**Funding:** This work was partially supported by the
National Heart, Lung and Blood Institute's
Framingham Heart Study (Contracts N01-HC-
25195 and HHSN2682015000001I) and by grants
R01HL131532 (SC), R01HL134168 (SC); and 2-
K24-HL04334. Beatrice von Jeinsen was supported
by the German Heart Foundation / German
Foundation of Heart Research. Gary F. Mitchell was
supported by Cardiovascular Engineering., Inc; this
funder provided support in the form of salary for
Gary F. Mitchell and various employees of
Cardiovascular Engineering, Inc., who are involved
in analysis of hemodynamic data. The
aforementioned salary support is derived from the
various NIH grants that supported the study.
Cardiovascular Engineering, Inc., did not have any
additional role in the study design, data collection
and analysis, decision to publish, or preparation of
the manuscript outside of the work done through
these NIH grants. The specific role of this author is
articulated in the 'author contributions' section.
There was no additional external funding received
for this study.

**Competing interests:** Susan Cheng has received
consulting fees from Zogenix for work unrelated to
this manuscript. Gary F. Mitchell has the following
disclosures: a) grants: NIH, Novartis (both
significant); b) consulting: Novartis, Servier, Merck,
Bayer (all significant); and c) ownership:
Cardiovascular Engineering, Inc. (significant).
These affiliations do not alter our adherence to
PLOS ONE policies on sharing data and materials.

without hypertension or diabetes compared to obese individuals with hypertension and diabetes. Presence of hypertension was associated with higher LVEF, whereas people with diabetes had lower LVEF.

## Conclusions

Obesity, hypertension, and diabetes interact synergistically to influence cardiac remodeling. These findings may explain the markedly heightened risk of heart failure and cardiovascular disease when these factors co-cluster.

## Introduction

Obesity, diabetes mellitus, and arterial hypertension are important independent risk factors for heart failure [1–3]. Accordingly, distinct forms of obesity cardiomyopathy [4–8], diabetic cardiomyopathy [9–13], and hypertensive heart disease [14–17] have been well described.

Obesity increases cardiac output by increasing central and total blood volume and lowering peripheral resistance [4–6]. These hemodynamic changes are accompanied by an increase in left ventricular (LV) wall stress, leading to LV diastolic dysfunction and hypertrophy (LVH), and left atrial (LA) enlargement [4–6]. Additionally, metabolic and neurohormonal changes in obesity (e.g., increased levels of myocardial triglycerides and fatty acids) may lead to subclinical myocardial dysfunction, reflected by an impairment of LV strain-based measurements [5–7], while LV ejection fraction (LVEF) is preserved [4, 5].

The metabolic changes seen in diabetes (such as hyperinsulinemia and hyperglycemia) alter myocardial metabolism and promote myocardial inflammation, fibrosis, and cardiac remodeling [9–12]. Whereas LVEF is often preserved in diabetes, the altered metabolic milieu can contribute to chamber remodeling, LV hypertrophy and diastolic dysfunction, and subtle impairment of myocardial systolic function [9, 11, 12].

Myocardial hypertrophy occurs as a compensatory mechanism to pressure overload in hypertension [15–17]. Myocyte hypertrophy is associated with interstitial fibrosis, changes in cardiomyocyte metabolism, myocyte apoptosis, and microvascular dysfunction. These myocardial changes in hypertension manifest as pathological LV and LA remodeling accompanied by diastolic dysfunction, LVH, and subtle myocardial systolic dysfunction, while LVEF is initially preserved [14–17]. Thus, obesity, diabetes mellitus and arterial hypertension all cause LVH, but it is yet unclear how their conjoint presence may influence cardiac structure, function and chamber geometry [4–6, 8, 9, 13, 14, 18–27].

Obesity, diabetes, and hypertension often coexist and presumably their conjoint presence may be associated with an adverse impact on cardiac structural and functional remodeling. Prior investigations have evaluated the effects of presence of these conditions in pair-wise combinations and reported that presence of any two of these conditions seems to be additively associated with adverse cardiac remodeling [28–37], worse LV diastolic function [28, 29, 31, 37], and an impairment of LV strain-based measures [38–40] and LV long axis function [28], whereas LVEF is preserved typically [28, 29, 35]. These previous investigations did not analyze the joint effects when all three conditions are present concomitantly [28–36, 39, 40], were mostly conducted in smaller samples [28, 29, 32–35, 37–39], and only a few studies investigated their possible interactions [28, 35, 36].

In the present investigation, we compared the independent and conjoint associations of obesity, diabetes mellitus, and arterial hypertension with a comprehensive panel of

echocardiographic measures including measures of LV size and geometry, LV systolic and diastolic function, LA size and function. We hypothesized that obesity, diabetes, and hypertension interact synergistically (rather than additively) measured by interaction terms to influence cardiac remodeling in the community.

## Materials and methods

### Study sample

The design and selection criteria for the Framingham Heart Study (FHS) Offspring Study, the Third Generation Cohort and the minority Omni Cohort 1 have been described elsewhere [41–43]. The present investigation included participants from the FHS Offspring cohort who attended their eighth examination cycle (N = 3021; 2005–08), the FHS Omni 1 cohort who attended their third examination cycle (N = 298; 2005–08), and the FHS Third Generation Cohort who attend their first examination cycle (N = 4095; 2002–05). The study protocol was approved by the Institutional Review Board of the Boston University Medical Center and all study participants provided written informed consent.

The analytic methods, data, and study materials will not be made available to other researchers for purposes of reproducing the results or replicating the procedure.

There were, 7,414 eligible participants from the three cohorts. We excluded 93 individuals with prevalent heart failure and 1580 individuals due to missing data (see **S1 Fig**), resulting in a final sample of 5741 participants.

### Measurements of covariates

During their FHS examinations, participants provided a detailed medical history, and underwent phlebotomy (after an overnight fast) for the assessment of CVD risk factors including a standard lipid panel and renal function, and a cardiovascular-targeted physical examination that included standardized anthropometry and blood pressure measurements. The presence of arterial hypertension was defined as a systolic blood pressure ≥140mm Hg or a diastolic blood pressure ≥90 mm Hg or the current use of antihypertensive medications. We classified participants as having diabetes mellitus if they had a fasting blood glucose concentration ≥126 mg/dL or if they were treated with any hypoglycemic medication. The presence of normal, overweight and obesity was defined based on the participant's body mass index (BMI; normal: BMI <25kg/m$^2$, overweight: 25kg/m$^2$ ≤BMI<30kg/m$^2$, obese: BMI ≥30kg/m$^2$).

Current smoking was defined as having smoked cigarettes regularly during the year antedating the FHS examination. Details of methods and criteria of measurement of all covariates have been published previously [44].

### Measurement of echocardiographic variables

All attendees underwent routine transthoracic echocardiography based on a standardized protocol on an HP Sonos 5500 ultrasound machine (Phillips Medical Systems, Andover, MA). Based on the recommendations of the American Society of Echocardiography (ASE), digitized images were obtained and measured offline. Digital images were used to measure LV end-systolic (LVSD) and end-diastolic (LVDD) dimensions as well as left atrial end-systolic (peak) dimension (LASD) and left atrial end-diastolic dimension (LADD). End-diastolic thicknesses of the LV posterior wall and the LV septum were summated to yield LV wall thickness (LVWT) [45]. Relative wall thickness (RWT) was calculated as equal to (LVWT)/LVDD) [46].

We estimated LV mass (LVM) using the method by Devereux et al, as follows: LVM = (0.8* (1.04(LVDD + septal wall thickness + posterior wall thickness)$^3$ –LVDD$^3$)+0.6) [47]. Because

LVM indexed to body surface area ($LVMI_{BSA}$) is used to classify LV geometry, but such indexation may underestimate LVH in obese individuals, we indexed LVM to height ($LVMI_{height}$) and to height ^2,7 ($LVMI_{height^{2.7}}$) additionally [36, 46].

We defined LV geometry thus: Normal LV geometry as $LVMI_{BSA} \leq 95$ g/m$^2$ in women or $LVMI_{BSA} \leq 115$ g/m$^2$ in men and RWT $\leq 0.42$. Concentric remodeling was defined as $LVMI_{BSA} \leq 95$ g/m$^2$ in women or $LVMI_{BSA} \leq 115$ g/m$^2$ in men and RWT $> 0.42$. LVH was defined as $LVMI_{BSA} > 95$ g/m$^2$ in women or $LVMI_{BSA} > 115$ g/m$^2$ in men. In the presence of LVH, eccentric LVH was defined as RWT $\leq 0.42$ and concentric LVH as RWT $> 0.42$ [46].

LVEF was calculated based on the Teichholz formula [48], which closely approximates that estimated by quantitative two-dimensional methods in the FHS laboratory [49]. We measured mitral annular plane systolic excursion (MAPSE), as recommended [50], by measuring the systolic excursion of the mitral annulus from its lowest point at end-diastole to its highest point at the time of aortic valve closure at the lateral side of the mitral valve annulus in the apical four-chamber view.

The speckle-tracking analyses of LV function was performed using an offline analysis program (2D Cardiac Performance Analysis version 1.1, TomTec Imaging Systems GmbH, Unterschleißheim, Germany) according to a standardized protocol that has been described in detail previously [51]. In summary, global longitudinal strain (GLS) was assessed in the apical two and apical four chamber views, and global circumferential strain (GCS) was assessed in the short axis view.

For the measurement of LV diastolic function, we derived early transmitral flow velocity (E), and the early systolic mitral annulus velocity (E', using tissue Doppler imaging at the lateral mitral annulus) to calculate the E/e' ratio [52].

We obtained maximum and minimum volumetric measurements of the LA (LAmax and LAmin) from apical two and four-chamber views in offline analysis of digital images using the recommended Simpson's biplane summation of disks method on a Digisonics DigiView System Software (ver. 3.7.9.3, Digisonics, Houston, TX). We calculated left atrial emptying fraction (LAEF) as ([LAmax—LAmin]/LAmax)*100 as previously published [46, 53]. LAmax and LAmin measurements were not available for the Third Generation Cohort, but left atrial fractional shortening (LAFS) was available for all participants (LAFS was calculated as [LASD-LADD]/LASD*100). Therefore, we used a linear regression model for the Offspring cohort, adjusting age, smoking, BMI, diabetes, systolic blood pressure, diastolic blood pressure, antihypertensive medication, heart rate, creatinine, high-density lipoprotein, low-density lipoprotein, and log(triglycerides), stratified by sex, with LAEF as the dependent variable, and LAFS, LAFS squared, and LAFS cubed as independent variables. Based on the resulting regression equations, we imputed LAEF values for all cohorts.

Measurements of the echocardiographic variables evaluated in this investigation are summarized and depicted in **Fig 1**. Data on inter-observer correlations have been previously published [51, 53, 54].

## Statistical analyses

We assessed baseline characteristics for the entire study sample and separately for each BMI category. We used natural-logarithm transforms of E/e', LASD, LVM, LVMI-BSA, LVMI-height, LVEF, LVWT, and RWT in order to normalize their skewed distributions.

There were twelve strata when the three BMI categories were cross-classified by diabetes (yes/no) and hypertension (yes/no) status. Using multivariable generalized linear models, we estimated least square means of LVM, LASD, LVEF, GLS, E/e', and LAEF (dependent variables) according to BMI category (normal, overweight and obese), hypertension (yes/no)

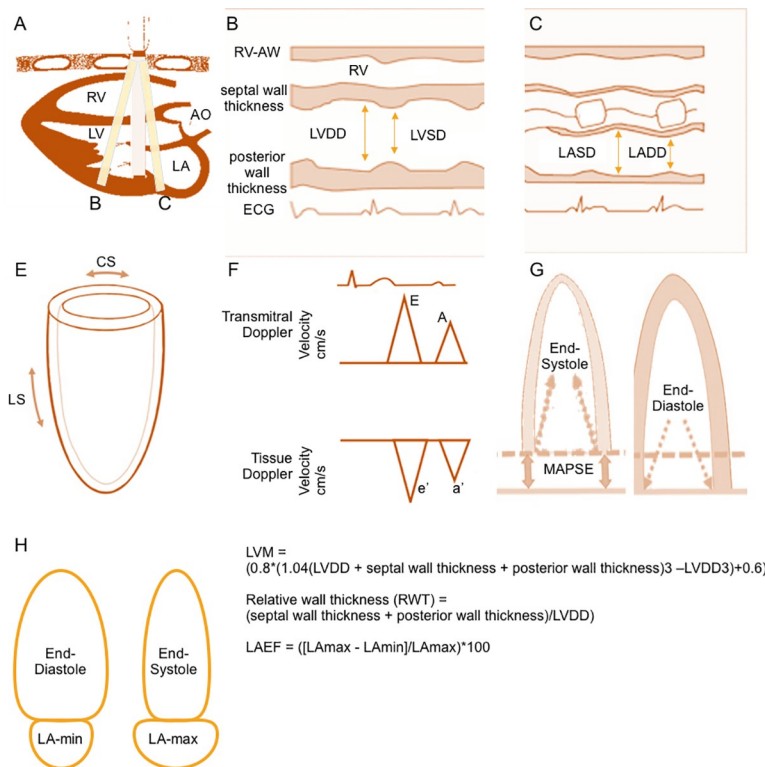

**Fig 1. Schematic presentation of the measurement of the investigated echocardiographic measures. Panel A** shows a 2D parasternal long axis view, lines B and C indicate the position of the m-mode. The respective m-mode view is shown in **panels B and C**. Left ventricular end-diastolic dimension (LVDD) and left ventricular end-systolic dimension (LVSD), posterior wall thickness, and septal wall thickness are measured as shown in **panel B**. Left ventricular mass (LVM), relative wall thickness (RWT) and left ventricular ejection fraction (LVEF) are calculated based on these measures. Left atrial end-systolic dimension (LASD) and left atrial end-diastolic dimension (LADD) are measured as shown in **panel C**. **Panel E** depicts the basic directions in which circumferential strain (CS) and longitudinal strain (LS) are measured. **Panel F** shows the measurements of transmitral Doppler (E and A) and tissue Doppler (e' and a') velocities. We have measured early systolic mitral annulus velocity using tissue Doppler imaging at the lateral mitral annulus. E/e' ratio is calculated based on these measurements. Mitral annular plane systolic excursion (MAPSE) was measured by the systolic excursion of the mitral annulus from its lowest point at end-diastole (**panel G**, right) to its highest point at the time of aortic valve closure (**panel G**, left) at the lateral side of the mitral valve annulus in the apical four-chamber view. Maximum and minimum volumetric measurements of the LA (LAmax and LAmin) from apical two and four-chamber views were taken at end-diastole for LAmin (**panel H**, left) and at end-systole for LAmax (**panel H**, right). Left atrial emptying fraction (LAEF) is calculated based on these measurements.

status, and diabetes (yes/no) status (independent exposure variables) separately and jointly. All models were adjusted for age, sex, and cohort status. We used a Bonferroni-corrected p value $<0.003$ ($= 0.05/(3*6)$) because we investigated 6 echocardiographic variables in relation to 3 different groups. We incorporated cross-product interaction terms for BMI category, hypertension status, and diabetes status to examine potential effect modification of associations (with the dependent variables) of each exposure individually by the conjoint presence of the other exposures. For the interaction terms, we used a p-value threshold of $p<0.05$ to indicate statistical significance.

We generated frequencies of normal LV geometry, concentric remodeling, eccentric LVH, and concentric LVH by BMI category, hypertension and diabetes status groupings.

In secondary analyses, we used regression models to estimate least square means of $LVMI_{BSA}$, $LVMI_{height}$, $LVMI_{height^{2.7}}$, LVWT, RWT, MAPSE, and GCS as additional measures of cardiac mass, wall thickness, and LV systolic and diastolic function, respectively. A less conservative p-value threshold of $<0.05$ was used for these exploratory analyses.

We calculated variance inflation factor values for all models, but we did not observe major collinearity. All analyses were performed using SAS version 9.4 (SAS, Cary, NC).

## Results

### Baseline characteristics

The clinical characteristics of our middle-aged sample (mean age 50 years, 54.6% women) are presented in **Table 1**. About 38% of the participants were overweight and 23% were obese,

**Table 1. Baseline characteristics of study sample, stratified by BMI category.**

| | Overall (n = 5741) | Normal weight (n = 2229) | Overweight (n = 2186) | Obese (n = 1326) |
|---|---|---|---|---|
| **Cardiovascular risk factors** | | | | |
| Age, years | 50 (15) | 47 (15) | 51 (15) | 52 (14) |
| Female, % | 54.6 | 69.7 | 41.4 | 51.1 |
| Height, cm | 169.1 (9.6) | 167.7 (9.1) | 170.7 (9.7) | 168.9 (9.7) |
| Weight, kg | 77 (17) | 63 (9) | 80 (10) | 98 (15) |
| Body mass index, kg/m$^2$ | 27.0 (5.1) | 22.4 (1.8) | 27.3 (1.4) | 34.1 (4.0) |
| Systolic blood pressure, mm Hg | 121 (16) | 115 (16) | 123 (16) | 125 (15) |
| Diastolic blood pressure, mm HG | 74 (10) | 72 (9) | 76 (9) | 77 (10) |
| Heart rate, bpm | 59 (9) | 59 (9) | 59 (9) | 61 (9) |
| No Smoking, % | 70 | 67 | 71 | 72 |
| Active Smoking, % | 17 | 18 | 17 | 16 |
| Former Smoking, % | 14 | 15 | 12 | 13 |
| Hypertension treatment, % | 22 | 12 | 24 | 35 |
| Prevalent CVD, % | 5 | 3 | 6 | 8 |
| Serum creatinine, mg/100ml | 0.84 (0.22) | 0.79 (0.16) | 0.87 (0.23) | 0.84 (0.26) |
| Total cholesterol, mg/100ml | 188 (35) | 185 (34) | 191 (36) | 190 (36) |
| High density lipoprotein, mg/100ml | 56 (17) | 63 (18) | 53 (15) | 50 (14) |
| Low density lipoprotein, mg/100ml | 110 (32) | 104 (31) | 114 (32) | 113 (32) |
| Triglycerides, mg/100ml | 109 (58) | 87 (45) | 116 (60) | 132 (63) |
| **Category** | | | | |
| No hypertension, no diabetes, N (%) | 3936 (68.6%) | 1838 (82.5%) | 1436 (65.7%) | 662 (49.9%) |
| Hypertension, no diabetes, N (%) | 1496 (26.1%) | 346 (15.5%) | 645 (29.5%) | 505 (38.1%) |
| Diabetes, no hypertension N (%) | 85 (1.5%) | 17 (0.8%) | 30 (1.4%) | 38 (2.9%) |
| Hypertension and diabetes, N (%) | 224 (3.9%) | 28 (1.3%) | 75 (3.4%) | 121 (9.1%) |
| **Echocardiographic indices** | | | | |
| Left ventricular mass, g | 161.1 (44.1) | 139.2 (35.1) | 170.2 (42.3) | 182.8 (44.5) |
| Left ventricular mass indexed by height, g/m | 94.7 (23.3) | 82.6 (18.2) | 99.2 (22.0) | 107.7 (23.4) |
| Left ventricular mass indexed by body surface area, g/m$^2$ | 84.2 (17.5) | 80.8 (16.0) | 87.0 (18.2) | 85.3 (17.7) |
| Left ventricular wall thickness, cm | 1.84 (0.26) | 1.72 (0.22) | 1.89 (0.24) | 1.97 (0.26) |
| Relative wall thickness | 0.38 (0.05) | 0.36 (0.05) | 0.38 (0.05) | 0.39 (0.06) |
| Left atrial systolic dimension, cm | 3.77 (0.51) | 3.48 (0.42) | 3.87 (0.45) | 4.09 (0.47) |
| Left ventricular ejection fraction, % | 66 (6) | 66 (6) | 66 (6) | 66 (6) |
| Mitral annular plane systolic excursion, cm | 1.57 (0.23) | 1.57 (0.22) | 1.57 (0.23) | 1.57 (0.23) |
| Global longitudinal strain, % | -20.3 (3.1) | -21.1 (3.1) | -19.9 (2.9) | -19.5 (3.2) |
| Global circumferential strain, % | -29.8 (5.3) | -29.6 (5.4) | -29.8 (5.2) | -30.0 (5.2) |
| E/e' | 6.3 (1.9) | 5.9 (1.7) | 6.3 (1.9) | 6.9 (2.0) |
| Left atrial emptying fraction, % | 48.1 (2.1) | 48.3 (2.0) | 48.0 (2.1) | 47.9 (2.2) |

Data are shown as means (standard deviation) for continuous variables and as percentage for categorical variables. Prevalent CVD = prevalent cardiovascular disease.

26% had hypertension but no diabetes, 2% had diabetes but no hypertension, and 4% had both hypertension and diabetes. In about 3% of the individuals, diabetes, hypertension and overweight/obesity co-clustered. Most participants with hypertension or diabetes were overweight or obese.

## Association of body mass index, hypertension, and diabetes with cardiac structural remodeling

When modeled individually, overweight/obese participants and individuals with hypertension or diabetes had higher LVM, higher LASD (**Table 2**), and, in secondary analysis, higher LVMI$_{height}$, LVMI$_{height^{2.7}}$, LVWT, and RWT compared to those with normal body mass index, and participants without hypertension or diabetes, respectively (**S1 Table**). Individuals with either hypertension or diabetes had higher LVMI$_{BSA}$ compared to individuals without hypertension or diabetes, but LVMI$_{BSA}$ did not differ between normal weight, overweight and obese individuals.

When we included BMI category, hypertension and diabetes status jointly in a model, we observed significant three-way multiplicative statistical interactions for LVM, and, in secondary analysis, for LVMI$_{height}$, LVMI$_{height^{2.7}}$, LVMI$_{BSA}$, LVWT, and RWT (**Table 3, S2 Table**).

We observed a pattern of increasing LVM values across the three BMI categories, for all combinations of hypertension and diabetes status. The increase in LVM between those with normal BMI and obese individuals was most pronounced among individuals with diabetes

**Table 2. Least square means of echocardiographic parameters by BMI category, hypertension status, and diabetes status (modeled separately), adjusted for age, sex, and cohort.**

| Trait | A) BMI Category | | | |
|---|---|---|---|---|
| | Normal Weight (39%) | Over-weight (38%) | Obese (23%) | P Value |
| **LVM, g** | 142.6 | 157.8 | 174.7 | **<0.0001** |
| **LASD, cm** | 3.53 | 3.78 | 4.03 | **<0.0001** |
| **LVEF, %** | 65.64 | 65.60 | 65.75 | 0.73 |
| **GLS, %** | -20.9 | -20.1 | -19.5 | **<0.0001** |
| **E/e'** | 5.71 | 6.07 | 6.56 | **<0.0001** |
| **LAEF, %** | 48.1 | 48.0 | 48.0 | 0.20 |

| Trait | B) Hypertension Status | | | C) Diabetes Status | | |
|---|---|---|---|---|---|---|
| | No HTN (70%) | HTN (30%) | P Value | No DM (95%) | DM (5%) | P Value |
| **LVM, g** | 151.0 | 166.0 | **<0.0001** | 154.5 | 169.9 | **<0.0001** |
| **LASD, cm** | 3.69 | 3.86 | **<0.0001** | 3.72 | 3.96 | **<0.0001** |
| **LVEF, %** | 65.36 | 66.34 | **<0.0001** | 65.70 | 64.84 | 0.0087 |
| **GLS, %** | -20.5 | -19.7 | **<0.0001** | -20.3 | -19.5 | **<0.0001** |
| **E/e'** | 5.86 | 6.46 | **<0.0001** | 6.00 | 6.76 | **<0.0001** |
| **LAEF, %** | 48.1 | 47.9 | **0.0005** | 48.1 | 47.3 | **<0.0001** |

Least squares means of left ventricular mass (LVM), left atrial systolic dimension (LASD), left ventricular ejection fraction (LVEF), global longitudinal strain (GLS), E/e', and left atrial emptying fraction (LAEF) according to A) body mass index (BMI) category (normal weight: BMI < 25kg/m$^2$, overweight: 25kg/m$^2$ ≤ BMI < 30kg/m$^2$, obese: BMI ≥ 30kg/m$^2$), B) hypertension status (HTN), C) and diabetes status (DM). All models are adjusted for cohort, age, and sex. We used a Bonferroni-corrected p<0.003 (= 0.05/(3*6)) because we investigated 6 echocardiographic variables in relation to 3 different groups.

**Table 3. Least square means of echocardiographic traits by BMI category, hypertension status, and diabetes status (modeled jointly), adjusted for age, sex, and cohort.**

| Echo Parameter | BMI Category | Healthy | DM, no HTN | HTN, no DM | HTN and DM | P-Value for 3-way interaction |
|---|---|---|---|---|---|---|
| **LVM (g)** | Normal | 140.7 | 141.5 | 150.7 | 140.7 | **0.0042** |
| | Overweight | 155.1 | 160.4 | 163.2 | 169.7 | |
| | Obese | 168.6 | 193.5 | 179.8 | 187.3 | |
| **LASD, cm** | Normal | 3.51 | 3.50 | 3.60 | 3.69 | 0.31 |
| | Overweight | 3.76 | 3.79 | 3.82 | 3.97 | |
| | Obese | 3.98 | 4.21 | 4.06 | 4.18 | |
| **LVEF, %** | Normal | 65.43 | 65.76 | 66.84 | 61.96 | **0.0002** |
| | Overweight | 65.35 | 66.26 | 66.36 | 64.21 | |
| | Obese | 65.34 | 63.16 | 66.40 | 66.46 | |
| **GLS, %** | Normal | -21.1 | -17.7 | -20.3 | -20.7 | **0.0005** |
| | Overweight | -20.2 | -19.3 | -19.7 | -19.1 | |
| | Obese | -19.7 | -19.3 | -19.1 | -19.2 | |
| **E/e'** | Normal | 5.60 | 5.77 | 6.09 | 6.41 | 0.25 |
| | Overweight | 5.92 | 6.22 | 6.39 | 6.91 | |
| | Obese | 6.36 | 7.30 | 6.68 | 7.23 | |
| **LAEF, %** | Normal | 48.1 | 47.1 | 48.0 | 48.5 | **0.0006** |
| | Overweight | 48.1 | 47.6 | 47.9 | 46.8 | |
| | Obese | 48.2 | 46.6 | 47.8 | 47.6 | |

Least squares means of left ventricular mass (LVM), left atrial systolic dimension (LASD), left ventricular ejection fraction (LVEF), global longitudinal strain (GLS), E/e', and left atrial emptying fraction (LAEF) according to body mass index (BMI) category (normal weight: BMI $< 25 kg/m^2$, overweight: $25 kg/m^2 \leq$ BMI $< 30 kg/m^2$, obese: BMI $\geq 30 kg/m^2$), hypertension status (HTN), and diabetes status (DM). All models are adjusted for cohort, age, and sex.

P-Value $<0.05$ was considered significant for interaction terms.

(37% increase) and with both diabetes and hypertension (33% increase) (**Fig 2A**). Within each BMI category, hypertension was associated with a similar increase of LVM of ~6%, while the increase in LVM seen in individuals with diabetes was more pronounced in obese (15%) versus in individuals with normal BMI (1%). We observed similar patterns in secondary analyses for LVMI$_{height}$, LVMI$_{height^2.7}$, LVWT and RWT (**S1A, S1C and S1D Fig**). In contrast, there was no increase of LVMI$_{BSA}$ with BMI category (**S1B Fig**).

Similar to LVM, LASD increased from normal BMI to overweight and to obese individuals for all combinations of diabetes and hypertension, with the strongest increase among those with diabetes (20%) (**Fig 2B**), but a three-way interaction term was statistically nonsignificant (consistent with additive effects).

## Association of body mass index, hypertension, and diabetes with cardiac function

When analyzed separately, overweight/obese participants and those with hypertension or diabetes had higher (presumably unfavorable) values of GLS and E/e' compared to those with normal body mass index, and those without hypertension or diabetes, respectively (**Table 2**). Individuals with hypertension or diabetes had lower values of LAEF compared with individuals without hypertension or diabetes, respectively. There was no difference in LAEF across the three BMI categories.

Individuals with hypertension had higher LVEF and, in secondary analysis, lower (more negative) GCS values (both presumably favorable effects) compared to individuals without hypertension. In contrast, individuals with diabetes had higher (less negative values) GCS

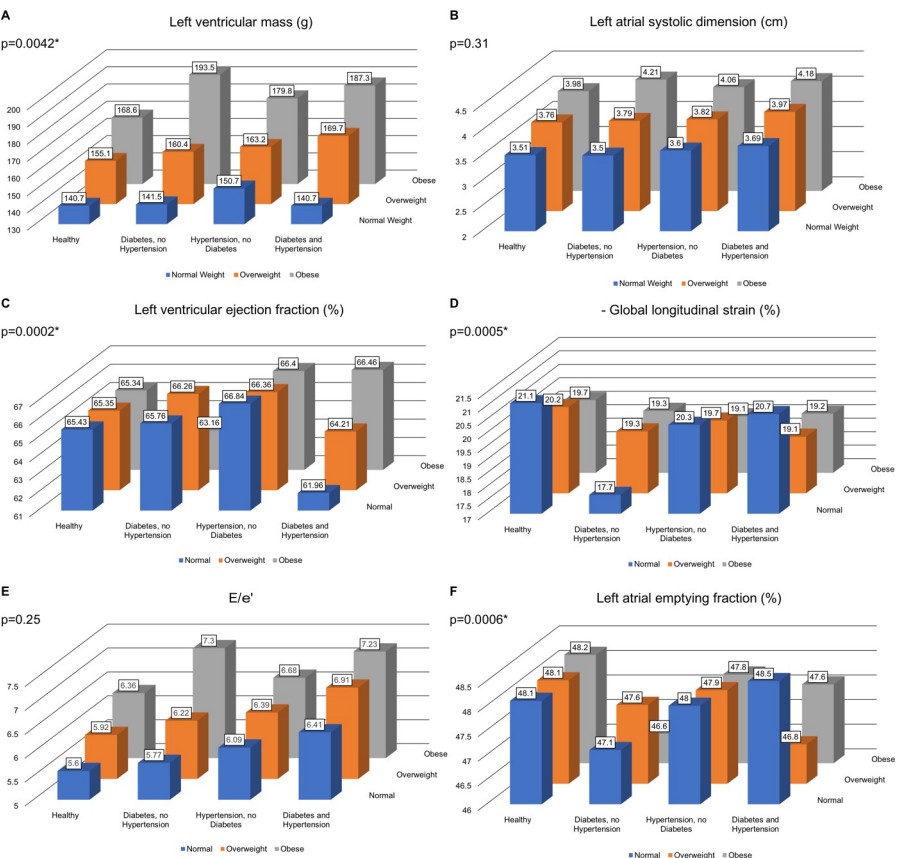

**Fig 2. Means of echocardiographic measures.** Least squares means of left ventricular mass (A), left atrial systolic dimension (B), left ventricular ejection fraction (C), negative global longitudinal strain (D), E/e' (E), and left atrial emptying fraction (F) according to cross-classified body mass index (normal weight: BMI < 25kg/m$^2$, overweight: 25kg/m$^2 \leq$ BMI < 30kg/m$^2$, obese: BMI $\geq$ 30kg/m$^{2)}$) hypertension, and diabetes status categories. All models are adjusted for cohort, age and sex. P values correspond to 3-way interactions.

values (**S1 Table**) compared to individuals without diabetes. Higher BMI was accompanied by an increase in MAPSE in secondary analysis (presumably a favorable effect) whereas individuals with hypertension or diabetes had lower MAPSE (**S1 Table**). We observed statistically significant three-way interactions between BMI categories, hypertension, and diabetes status for LVEF, GLS, and LAEF (**Table 3**) in the joint models.

LVEF (**Fig 2C**) was higher in normal weight, overweight, and obese individuals with hypertension compared to normal weight, overweight, and obese individuals without hypertension (+2% increment in LVEF for all three groups). Normal and overweight individuals with diabetes had higher LVEF compared to normal and overweight individuals without diabetes (+1% increment in LVEF). Obese individuals with hypertension and diabetes had higher LVEF than individuals with a normal BMI who had hypertension and diabetes (+7% increment in LVEF). In contrast, LVEF was lower in obese individuals with diabetes compared to their normal BMI counterparts (4% decrement in LVEF).

GLS (**Fig 2D, GLS is depicted as -GLS**) values were higher (less negative, i.e., presumably unfavorable effects) in individuals with diabetes and/or hypertension in all BMI categories (+2% to +16% increase) and in overweight or obese individuals compared to individuals with normal BMI in individuals with hypertension or those with hypertension and diabetes (+6% to +7% increase, respectively).

We investigated MAPSE and GCS (**S2E and S2F Fig, GCS is depicted as -GCS**) as additional measures of LV long axis and circumferential function in secondary analyses. MAPSE was lower in individuals with diabetes and/or hypertension in all BMI categories (0 to -3% decrement) compared to individuals without diabetes and/or hypertension. However, MAPSE was higher in obese individuals compared to that in individuals with normal BMI irrespective of their diabetes or hypertension status (2% to 3% increment).

GCS values were higher (less negative, i.e., presumably unfavorable) in obese individuals compared to individuals with normal BMI with diabetes and/or hypertension (1% to 2% worse) and in overweight participants with diabetes and hypertension compared to their counterparts with normal BMI (5% worse). In contrast, GCS values were lower (more negative, i.e., presumably beneficial) in individuals with hypertension compared to individuals without hypertension in all three BMI categories (-2% to -3% lower/better).

E/e' (**Fig 2E**) was 3% higher in participants with normal BMI with diabetes compared to participants with normal BMI without diabetes, and it was 15% higher in obese subjects with diabetes compared to obese subjects without diabetes. Individuals with hypertension had 5–8% higher E/e' than normotensive individuals in all BMI categories, and individuals with both diabetes and hypertension had 14–17% higher E/e' than individuals without either condition (in all BMI groups). The highest increase in E/e' was observed in obese individuals with diabetes (27% increase relative to those with normal BMI).

While there were no differences in LAEF between individuals with normal BMI, overweight or obesity (**Fig 2F**), individuals with hypertension and/or diabetes had lower adjusted mean values of LAEF compared to individuals without hypertension and/or diabetes. We observed the lowest values of LAEF in obese participants with diabetes but without hypertension, and in overweight participants with diabetes and hypertension.

### Association of body mass index, hypertension, and diabetes with LV geometric patterns

The prevalence of normal LV geometry decreased and that of concentric remodeling, eccentric LVH, and concentric LVH increased among those with hypertension and/or diabetes regardless of their BMI category (**Fig 3**, **S3 Table**). Only about 50% of subjects with both hypertension and diabetes had normal LV geometry. Overweight and obesity were associated with an increase in concentric remodeling in individuals without hypertension or diabetes. With increasing BMI, there were only small changes in the prevalence of LV geometric patterns in individuals with hypertension. In the presence of diabetes, the presence of overweight and obesity shifted the distribution of LV geometry to higher frequencies of concentric remodeling and concentric LVH (relative to those without diabetes). In participants with both diabetes and hypertension, overweight and obesity led to only a small change of the frequency of abnormal geometry generally but increased the frequency of concentric LVH.

## Discussion

We investigated conjoint influences of obesity, hypertension, and diabetes on cardiac structure and function, and chamber geometry in a large community-based sample free of heart failure.

### Principal findings

Our principal findings are four-fold, as summarized in **Table 4**.

First, we observed that BMI category, hypertension, and diabetes were each associated with cardiac remodeling and altered LV geometry. Most of these associations reflected unfavorable

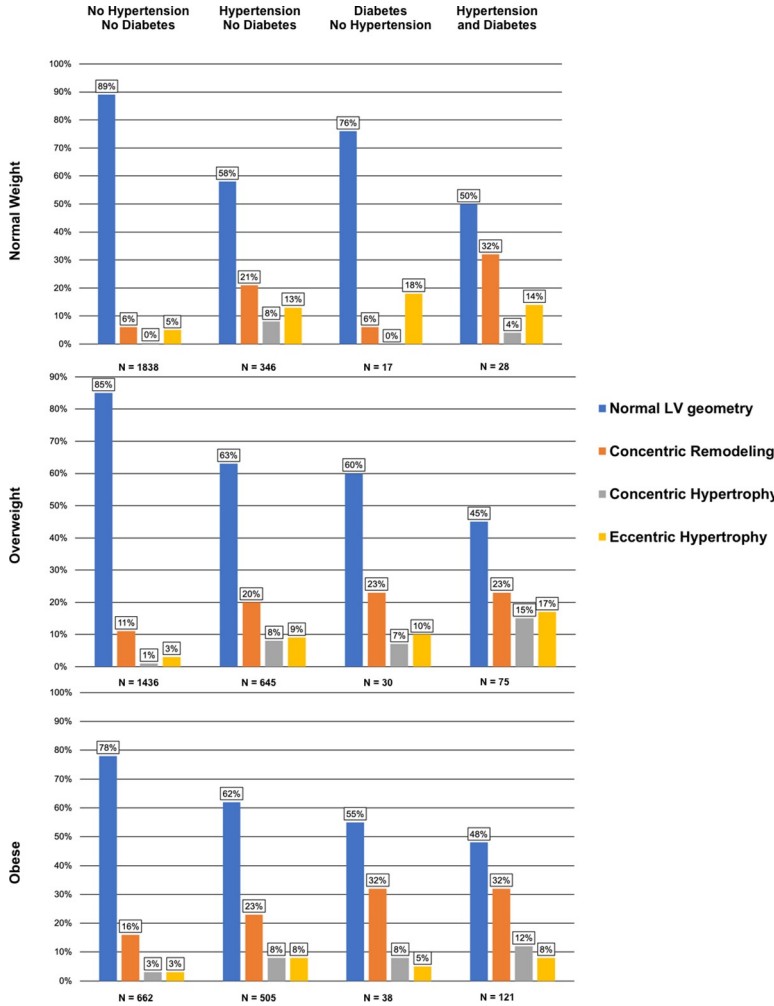

**Fig 3. Distribution of left ventricular geometry.** Frequencies (percentages) of left ventricular geometry (LV) patterns by body mass index (BMI) category (normal weight: BMI < 25kg/m², overweight: 25kg/m² ≤ BMI < 30kg/m², obese: BMI ≥ 30kg/m²), hypertension status, and diabetes status.

consequences, although we observed associations between hypertension and higher LVEF and better GCS, and between higher BMI and greater MAPSE (see below).

Second, when modeled jointly, we observed significant three-way statistical interactions between BMI category, hypertension, and diabetes for LVM, LVEF, GLS, and LAEF; and in

**Table 4. Summary of associations.**

|  | Three-way interaction + | | | | Additive effects | | | |
|---|---|---|---|---|---|---|---|---|
|  | **LVM** | **LVEF** | **LAEF** | **GLS** | **LASD** | **E/e'** | **MAPSE** | **GCS** |
| Hypertension | ↑ | ↑ | ↓ | ↑ | ↑ | ↑ | ↓ | ↑ |
| Obesity | ↑ | → | → | ↑ | ↑ | ↑ | ↑ | → |
| Diabetes | ↑ | ↑ | ↓ | ↑ | ↑ | ↑ | ↓ | ↓ |

↑ indicates values increase (become more positive/less negative in case of GLS and GCS)

→ indicates no association

↓ indicates values decrease (become less positive/more negative in case of GLS and GCS)

secondary analyses, for LVMI, LVWT, and RWT. The strengths of the unfavorable associations of these three risk factors (overweight/obesity, hypertension and diabetes) with LVM, GLS, and LAEF suggest that these influences are synergistic. In contrast, for select echocardiographic measures, the associations of these risk factors were opposite in directionality—for instance, individuals with hypertension tended to have higher LVEF compared to their non-hypertensive counterparts (BMI being held constant) whereas obese individuals had a lower LVEF compared to those with a normal BMI (blood pressure being held constant).

Third, there were no statistically significant synergistic interactions for these three risk factors in relation to their associations with LASD or E/e' in primary analyses, and for their relations to MAPSE and GCS in secondary analyses. There were unfavorable additive effects of BMI, diabetes and hypertension on LASD and E/e'. For select echocardiographic measures the associations of these three risk factors varied in directionality. For instance, obesity was positively associated with MAPSE, whereas diabetes or hypertension were negatively related to long axis LV function. Likewise, hypertension was associated with better (more negative) GCS, whereas relations for obesity or diabetes were opposite in direction.

Fourth, we observed a lower prevalence of normal LV geometry in obese individuals and in individuals with hypertension or diabetes; normal LV geometry was least common in participants with both hypertension and diabetes (**Fig 3**) suggesting synergistic effects.

## Individual associations of body mass index, diabetes and hypertension with cardiac remodeling

Our findings that a higher BMI category and the presence of hypertension or diabetes each is associated with adverse LV and LA remodeling have been reviewed extensively in the literature along with delineation of the underlying mechanisms [4–17]. The observation, that a higher BMI category was not associated with $LVMI_{BSA}$ but was associated with greater LVM, $LVMI_{height}$ and $LVMI_{height^{2.7}}$, underscore previous observations that indexation of LVM by body surface area may mask the adverse impact of obesity on cardiac mass and hypertrophy [55].

We observed, that adjusted mean values of GCS were more negative (presumably favorable effects) in subjects with hypertension, but higher (less negative, i.e., unfavorable) in subjects with diabetes. There is evidence that in the early stages of myocardial response in hypertension, when GLS may be impaired, GCS is augmented to maintain LVEF [56]. It is possible that the subtle myocardial dysfunction in our study sample was more pronounced in participants with diabetes compared to individuals with hypertension, leading to augmented GCS in individuals with hypertension but worse GLS and GCS in those with diabetes. In our investigation adjusted mean values of LVEF were higher in individuals with hypertension, which has been described previously [57, 58]. One possible explanation for the increase in LVEF might be an increase in ventricular torsion and/or LV wall thickening in individuals with hypertension [58].

Presumably unfavorable associations between diabetes [28] and hypertension [28, 59] with MAPSE have been described previously [48]. We observed that MAPSE was higher (presumably a favorable effect) in overweight/obese subjects. So far, only unfavorable associations between MAPSE and body weight have been described by three prior studies with small samples (n<120) [28, 60, 61], two of which investigated severely obese individuals with a mean BMI >40kg/m$^2$ [60, 61]. Therefore, previous studies may not be comparable to our present investigation of a larger community-based sample. A possible explanation for the favorable association of higher BMI with MAPSE observed in our investigation might be the increase cardiac work necessitated with higher body weight [4–6]. Additional studies including larger samples are warranted to confirm our findings and to investigate the physiological mechanisms underlying the observed associations.

## Conjoint associations of body mass index, diabetes and hypertension with cardiac structural remodeling

Previous investigators have reported additive and unfavorable associations of obesity and hypertension [30–33], obesity and diabetes [30–32, 37], and hypertension and diabetes [28–30, 37] with measures of cardiac remodeling. Two studies investigated possible interactions between hypertension and obesity on LVM. A prior study of severely obese individuals reported an interaction [35], while a previous report from the FHS did not observe any interaction between obesity and hypertension in terms of their associations with LVM but that report excluded participants who were on antihypertensive medications and presumably had longstanding high blood pressure [36].

We observed significant three-way statistical interactions between BMI category, hypertension, and diabetes for LVM, LVMI and LVWT, which indicates that BMI, hypertension and diabetes have interrelated influences on LV remodeling. It is noteworthy that an overweight individual without diabetes or hypertension already has comparable LVM and LVWT as an individual with a normal BMI who has both hypertension and diabetes. Although a participant with hypertension and diabetes is usually considered to be at risk for heart failure, a similar risk (of heart failure) might be underappreciated in overweight but seemingly healthy individuals. The high values of LVM observed in obese individuals with diabetes emphasizes the higher risk of heart failure observed in such individuals, thereby might underscore the importance of weight loss and optimal management of diabetes in obese individuals. In fact, a recent echocardiographic study of patients with diabetes reported that those with diabetes but with low rates of hypertension and obesity remained at higher risk of cardiovascular mortality and hospitalization if they had increased LVMI [37].

## Conjoint associations of body mass index, diabetes, and hypertension with LV diastolic function

A previous study [31] of approximately 2500 participants investigated the associations between BMI category and several echocardiographic markers including E/e' and E/A in the presence or absence of at least one component of the metabolic syndrome (including diabetes and hypertension). That study reported that obesity was associated with worse LV diastolic function in both metabolically healthy and unhealthy individuals, and that unhealthy participants had worse LV diastolic function compared to their counterparts with a normal BMI [31].

Similarly, we observed an increase in E/e' with increasing BMI category, with the increase being more pronounced in obese individuals with diabetes compared to obese individuals without diabetes. We observed higher values of E/e' for individuals with normal BMI with hypertension compared to individuals with normal BMI with diabetes. We did not observe any statistically significant interaction between BMI category, hypertension and diabetes for E/e'. Overall, our results suggest that there are additive associations of BMI category, hypertension and diabetes on LV diastolic function.

## Conjoint associations of body mass index, diabetes, and hypertension with LV systolic function

There are only few reports on the possible conjoint associations of obesity, hypertension, and diabetes with LVEF [28, 29, 35], GLS [38–40], GCS [38], and MAPSE [28]. These investigations reported unfavorable associations between different combinations of obesity, hypertension and diabetes, with GLS, GCS, and MAPSE, but not with LVEF [28, 29, 35, 37–40].

We observed a significant three-way multiplicative interaction for the associations of BMI category, hypertension, and diabetes with GLS, which suggests synergistic effects. Individuals with all three conditions or the combination of obesity with either diabetes or hypertension had the highest (less negative, i.e., unfavorable) mean adjusted values of GLS.

Hypertension was associated with better (more negative) mean GCS, but this association was attenuated by the presence of a higher BMI category or diabetes.

Overweight and obesity were associated with higher (favorable) mean MAPSE values compared to normal weight individuals in all subgroups (no additional risk factors, only hypertensive, only diabetes, and both, diabetes and hypertensive), but MAPSE was lower (unfavorable) in individuals with hypertension, diabetes, or both if BMI was held constant. Overall, these observations are consistent with a possible greater long axis LV shortening with higher BMI category, but attenuated shortening in the presence of concomitant hypertension and diabetes (on MAPSE). As mentioned above, additional studies are warranted to confirm this observation and to investigate potential physiological mechanisms."

## Conjoint associations of body mass index, diabetes and hypertension with LAEF

We observed lower adjusted mean values for LAEF in individuals with diabetes or hypertension compared to individuals without diabetes or hypertension, respectively. However, we did not observe any statistically significant associations between BMI category and LAEF. The inverse associations of hypertension and diabetes with LAEF seemed to be synergistic, although the absolute magnitude of the effect sizes (0–3%) were very modest and their clinical significance is unclear.

## Conjoint associations of body mass index, diabetes and hypertension with LV geometric patterns

In individuals without diabetes or hypertension, there was an overall low prevalence of abnormal LV geometry—we observed an increase in the prevalence of a concentric remodeling pattern with overweight and obesity (compared to normal BMI). There were relatively more overweight and obese individuals with eccentric compared to concentric LVH. Both, the presence of concentric and eccentric LVH have been reported in obese individuals with a tendency towards a higher prevalence of eccentric LVH. It has been suggested that concomitant hypertension might promote a change in the geometric pattern to concentric LVH [6, 8, 22].

Individuals with hypertension (compared to their non-hypertensive counterparts) had a relatively higher prevalence of eccentric compared to concentric LVH, irrespective of their BMI category. The question whether concentric or eccentric LVH is more frequent in individuals with hypertension has been discussed controversially and recent reviews report similar prevalence of concentric and eccentric LVH or even a slightly higher prevalence of eccentric LVH in hypertensive individuals. These reports emphasize the role of comorbidities to influence the pattern of LVH [14, 23].

Previous investigations reported an association between diabetes and abnormal LV geometry, but were inconsistent about the relative prevalence of concentric versus eccentric remodeling/LVH [18–20]. We observed that in participants with diabetes and a normal BMI/overweight, eccentric LVH was the predominant geometric pattern. Yet, in obese individuals with diabetes the prevalence of concentric LVH was higher. Thus, the BMI category may determine the pattern of LV geometry in people with diabetes.

Obese individuals with both diabetes and hypertension had a higher prevalence of concentric compared to eccentric LVH. It is possible, that the combination of volume overload

(obesity), metabolic changes (diabetes, obesity) and pressure overload (hypertension) synergistically contribute to the development of concentric LVH, which is known to most adversely impact overall survival [21].

## Strength and limitations

The strengths of our investigation include the large community-based sample, the evaluation of a comprehensive panel of novel and standard echocardiographic markers of cardiac remodeling and aortic stiffness, and the study of the synergistic interactions between BMI category, hypertension, and diabetes. However, several limitations of our approach must be considered. Although we included the racially diverse minority FHS Omni cohort in our sample, our overall study sample was comprised predominantly of whites of European ancestry with a low overall prevalence of diabetes. Future validation of our observations in large multiethnic cohorts is necessary, especially as the influence of race on LVH has been emphasized [18]. It must be noted that the mean differences in LVEF across the different strata were small and within the normal range. Our large sample size might have given us the possibility to detect very modest differences in echocardiographic measure of LV systolic and LA function that may be statistically significant, but these small differences might not be clinically meaningful. Due to the observational as well as cross-sectional design of our investigation, it is not possible to make causal interferences and we cannot exclude residual confounding by additional factors not adjusted for.

## Conclusions

In our large community-based sample, we observed synergistic interactions between a higher BMI category and the presence of hypertension or/and diabetes and their associations with adverse cardiac structural and functional remodeling. Additional studies of larger and multi-ethnic samples are warranted to confirm our findings and investigate whether these synergistic effects are also linked to an increased risk of clinical outcomes when these risk factors co-cluster.

## Supporting information

**S1 Fig. Flow diagram representing selection and exclusion process.** Abbreviations: CHF: congestive heart failure; Gen 3: Third Generation Framingham Cohort; Offspring: Framingham Offspring Study; Omni 2: Omni 2 Cohort.
(DOCX)

**S2 Fig.** Least squares means of left ventricular mass indexed by height (A), left ventricular mass indexed by body surface area (B), left ventricular wall thickness (C), relative wall thickness (D), mitral annular plane systolic excursion (E), and negative global circumferential strain (F) according to cross-classified body mass index (normal weight: BMI $< 25kg/m^2$, overweight: $25kg/m^2 \leq BMI < 30kg/m^2$, obese: BMI $\geq 30kg/m^{2)}$) hypertension, and diabetes status categories. All models are adjusted for cohort, age and sex. P values correspond to 3-way interactions.
(DOCX)

**S1 Table. Least square means of echocardiographic traits by BMI category, hypertension status and diabetes status (modeled separately) in secondary analysis, adjusted for age, sex, and cohort.** Least squares means of left ventricular mass index (LVMI) indexed by height (g/m), height^2.7, and body surface area (g/m$^2$), left ventricular wall thickness (LVWT), relative wall thickness (RWT), mitral annular plane systolic excursion (MAPSE), global

circumferential strain (GCS) according to body mass index (BMI) category (normal weight: BMI $< 25$kg/m$^2$, overweight: $25$kg/m$^2 \leq$ BMI $< 30$kg/m$^2$, obese: BMI $\geq 30$kg/m$^2$), hypertension status (HTN), and diabetes status (DM). All models are adjusted for cohort, age, sex. Bold print: $p < 0.05$, considered significant in secondary analysis.
(DOCX)

**S2 Table. Least square means of echocardiographic parameters stratified by BMI category, hypertension status and diabetes status (modeled jointly) in secondary analysis, adjusted for age, sex, and cohort.** Least squares means of left ventricular mass index (LVMI) indexed by height, height^2,7, and body surface area, left ventricular wall thickness (LVWT), relative wall thickness (RWT), mitral annular plane systolic excursion (MAPSE), and global circumferential strain (GCS) according to body mass index (BMI) category (normal weight: BMI $<$ $25$kg/m$^2$, overweight: $25$kg/m$^2 \leq$ BMI $< 30$kg/m$^2$, obese: BMI $\geq 30$kg/m$^2$), hypertension status (HTN), and diabetes status (DM). All models are adjusted for cohort, age, sex. P-Value $<0.05$ was considered significant for interaction terms.
(DOCX)

**S3 Table. Frequencies of left ventricular geometry category by BMI category, hypertension status and diabetes status.** Frequencies (percentage) of LV geometry category by BMI category (normal weight: BMI $< 25$kg/m$^2$, overweight: $25$kg/m$^2 \leq$ BMI $< 30$kg/m$^2$, obese: BMI $\geq 30$kg/m$^2$), hypertension status (HTN) and diabetes status (DM). Percentages represent frequency/row total.
(DOCX)

## Acknowledgments

We thank the Framingham Heart Study participants for making this research investigation possible.

## Author Contributions

**Conceptualization:** Beatrice von Jeinsen, Ramachandran S. Vasan, Vanessa Xanthakis.

**Data curation:** David D. McManus.

**Formal analysis:** Ramachandran S. Vasan, Vanessa Xanthakis.

**Funding acquisition:** Ramachandran S. Vasan, Gary F. Mitchell, Susan Cheng.

**Investigation:** Beatrice von Jeinsen, Gary F. Mitchell, Vanessa Xanthakis.

**Validation:** David D. McManus, Susan Cheng, Vanessa Xanthakis.

**Visualization:** Susan Cheng, Vanessa Xanthakis.

**Writing – original draft:** Beatrice von Jeinsen, Ramachandran S. Vasan, Vanessa Xanthakis.

**Writing – review & editing:** Ramachandran S. Vasan, David D. McManus, Gary F. Mitchell, Susan Cheng, Vanessa Xanthakis.

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
