## [Decision Letter · Decision Letter 0]

17 Sep 2020

PONE-D-20-14346

Joint influences of obesity, diabetes, and hypertension on indices of ventricular remodeling: findings from the community-based Framingham Heart Study

PLOS ONE

Dear Dr. Xanthakis,

Thank you for submitting your manuscript to PLOS ONE. After careful consideration, we feel that it has merit but does not fully meet PLOS ONE’s publication criteria as it currently stands. Therefore, we invite you to submit a revised version of the manuscript that addresses the points raised during the review process.

We look forward to receiving your revised manuscript.

Kind regards,

Elena Cavarretta, M.D., Ph.D.

Academic Editor

PLOS ONE

Journal Requirements:

2. Our internal editors have evaluated your manuscript and determined that it is within the scope of our 'Primary and Secondary Prevention of Cardiovascular Disease' Call for Papers.

This collection of papers is headed by a team of Guest Editors for PLOS ONE and will encompass a diverse range of research articles.

Additional information can be found on our announcement page: (https://collections.plos.org/s/prevention-cardiovascular).

If you would like your manuscript to be considered for this collection, please let us know in your cover letter and we will ensure that your paper is treated as if you were responding to this call. If you would prefer to remove your manuscript from collection consideration, please specify this in the cover letter.

'This work was partially supported by the National Heart, Lung and Blood Institute’s Framingham Heart Study (Contracts N01-HC-25195, HHSN268201500001I and 75N92019D00031) and by grants HL076784, G028321, HL070100, HL060040, HL080124, HL071039, HL077447, HL107385, HL126136, 2R01HL092577, 1R01HL128914, 1P50HL120163, R01HL131532, R01HL134168, R01HL143227, and 2-K24-HL04334. DDM was supported by grants U54HL143541, R01HL126911, R01HL137734, R01HL137794, R01HL135219, and R01HL141434 from the National Heart, Lung and Blood Institute and National Center for Complementary and Integrative Health (National Institute of Health), and Grant NSF-12-512 from the National Science Foundation. BVJ was supported by the German Heart Foundation / German Foundation of Heart Research. Dr. Vasan is supported by an Evans Scholar award and Jay and Louis Coffman Foundation from the Department of Medicine, Boston University School of Medicine.'

We note that one or more of the authors are employed by a commercial company: Cardiovascular Engineering, Inc

Reviewers' comments:

Reviewer's Responses to Questions

**Comments to the Author**

1. Is the manuscript technically sound, and do the data support the conclusions?

Reviewer #1: Yes

Reviewer #2: Yes

2. Has the statistical analysis been performed appropriately and rigorously? 

Reviewer #1: I Don't Know

Reviewer #2: Yes

3. Have the authors made all data underlying the findings in their manuscript fully available?

Reviewer #1: Yes

Reviewer #2: Yes

4. Is the manuscript presented in an intelligible fashion and written in standard English?

Reviewer #1: Yes

Reviewer #2: Yes

5. Review Comments to the Author

Reviewer #1: Von Jeinsen et al. utilized data from the Framingham study to investigate the impact of obesity, hypertension and diabetes on cardiac remodeling. English language is fine, but some punctuation typos are present. The paper is well written and the topic interesting.

I suggest providing an image (maybe more than one panel) depicting all echocardiographic measures taken, as not all readers might be familiar with the terms used and a graphical sketch may improve the readability of the manuscript.

Reviewer #2: Joint influences of obesity, diabetes, and hypertension on indices of ventricular remodeling: findings from the community-based Framingham Heart Study is a well written study based on 7414 participants in the Offspring and Omni Cohorts of The Framingham Study.

The synergistic effect of Hypertension, Obesity and diabetes have earlier been shown for each pair of exposures, but this study adds new knowledge by assessing the joint effect of all three factors on left ventricular structure and function. They add new knowledge by in addition to LVH also assessing left atrial size by systolic diameter, LV longitudinal strain, left atrial emptying fraction and E/e’ .

They assess the interaction by assessing the association for each parameter in healthy, those with diabetes only, hypertension only and those with both conditions stratified on normal, overweight and obese participants.

They do find an association increasing with increasing number of factors for LVM, LVEF and GLS but not for left atrial diameter or E/e’, the latter both markers of diastolic dysfunction which one would expect to be related to LVM.

That GLS is affected stronger than EF is expected as GLS is more sensitive to systolic dysfunction than EF.

Minor comments:

The common abbreviation of early filling velocity over Tissue Doppler early velocity, e prime is usually written E/e’.

Diabetes is often undiagnosed in 20-40% of subjects. Better to use HbA1c >48 mmol/mol (6.5%) to detect these subjects than glucose measurements if HbA1c is available.

In Table 1 To not describe former smoking often misses clear effects of smoking status.

According to current guidelines indexation is always advised. For LVM use of height¨2.7 is preferred as it has the strongest relation to subsequent outcome (ref ESC guidelines in hypertension ect)

As shown in the analyses and earlier shown in the early Framingham studies on LVH indexation by body surface area masks the effect of obesity on LVH. I would therefore drop analyses on LVM/BSA and LVM/h and focus on LVM/h¨^2.7.

As there was a significant interaction term HT*DMII*BMI, I would drop table 2 and focus on table 3.

Indexation should also be done for LASd with BSA. See Michael Stylidis Echocardiography. 2019 Mar;36(3):439-450.

If LAvolume is available this would be preferred instead of LASd, again indexed by BSA.

It would be appropriate also to mention interobserver data in the echocardiographic measurements.

As BMI, LVM, hypertension and diabetes all are correlated, it would strengthen the discussion to also report Variance inflation factor for these associations and a control of residuals to assure the GLM models used to calculate LS means are valid.

The increasing LVEF in subjects with hypertension could be due to concentric hypertrophy leaving the actual LV volume smaller. Then to maintain cardiac output EF has to increase to maintain stroke volume if heart rate does not increase.

The finding of a decreased LVEF for obese non hypertensive subjects could be due to the increase in circulating volume seen with obesity thought to be a cause of dilatation and heart failure associated with obesity. This could be tested in this study by assessing LVDd in these groups and see if obese non hypertensive are more dilated than normal non hypertensivs.

The finding of an increasing MAPSE for increasing BMI is interesting and although not significant the same trend is seen in stratified analysis over all four groups of health, only hypertensive, only diabetes and both. As the decrease in MAPSE over these groups is not significant in overall analysis this is also interesting.

I would suggest that the presentation and discussion focuses on the overall stratification in BMI groups and on healthy, only hypertensive, only diabetes and both and effects disappearing here but apparent in selective two way analysis can be discussed in relation to power or other possible reasons for the lack of overall effect.

6. PLOS authors have the option to publish the peer review history of their article (what does this mean?). If published, this will include your full peer review and any attached files.

Reviewer #1: No

Reviewer #2: **Yes: **Henrik Schirmer

---

## [Author Response · Author response to Decision Letter 0]

6 Nov 2020

Please see attached Cover letter and Response to Reviewers document.

---

## [Decision Letter · Decision Letter 1]

18 Nov 2020

Joint influences of obesity, diabetes, and hypertension on indices of ventricular remodeling: findings from the community-based Framingham Heart Study

PONE-D-20-14346R1

Dear Dr. Xanthakis,

We’re pleased to inform you that your manuscript has been judged scientifically suitable for publication and will be formally accepted for publication once it meets all outstanding technical requirements.

Kind regards,

Elena Cavarretta, M.D., Ph.D.

Academic Editor

PLOS ONE

Additional Editor Comments (optional):

Reviewers' comments:

Reviewer's Responses to Questions

**Comments to the Author**

1. If the authors have adequately addressed your comments raised in a previous round of review and you feel that this manuscript is now acceptable for publication, you may indicate that here to bypass the “Comments to the Author” section, enter your conflict of interest statement in the “Confidential to Editor” section, and submit your "Accept" recommendation.

Reviewer #1: All comments have been addressed

2. Is the manuscript technically sound, and do the data support the conclusions?

Reviewer #1: Yes

3. Has the statistical analysis been performed appropriately and rigorously? 

Reviewer #1: Yes

4. Have the authors made all data underlying the findings in their manuscript fully available?

Reviewer #1: Yes

5. Is the manuscript presented in an intelligible fashion and written in standard English?

Reviewer #1: Yes

6. Review Comments to the Author

Reviewer #1: The authors have addressed my concerns and I have no further comments or criticism. I particularly like the added figure, which I believe improved the readability of the manuscript to whom is not particularly familiar with echocardiography.

7. PLOS authors have the option to publish the peer review history of their article (what does this mean?). If published, this will include your full peer review and any attached files.

Reviewer #1: No

---

## [Editor Report · Acceptance letter]

1 Dec 2020

PONE-D-20-14346R1 

Joint influences of obesity, diabetes, and hypertension on indices of ventricular remodeling: findings from the community-based Framingham Heart Study 

Dear Dr. Xanthakis:

I'm pleased to inform you that your manuscript has been deemed suitable for publication in PLOS ONE. Congratulations! Your manuscript is now with our production department. 

Kind regards, 

on behalf of

Dr. Elena Cavarretta 

Academic Editor

PLOS ONE